# Distributed Energy Management for Ship-Integrated Energy System Considering Economic and Environmental Benefits

1st Yuxin Zhang
*Navigation College*
*Dalian Maritime University*
Dalian, China
liam_zhang@dlmu@edu.cn

2nd Qihe Shan
*Navigation College*
*Dalian Maritime University*
Dalian, China
tengfei@dlmu.edu.cn

3rd Haoran Liu
*Navigation College*
*Dalian Maritime University*
Dalian, China
lhr6@dlmu.edu.cn

4th Tieshan Li
*School of Automation Engineering*
*University of Electronic Science and Technology of China*
Chengdu, China
litieshan073@uestc.edu.cn

*Abstract*—To decrease the dependency of fuel-based resources in the shipping industry, the energy management problem is analyzed in this paper. Firstly, the development of shipboard energy system is reviewed from radiation pattern, ring pattern, two-end pattern to the ship-integrated energy system. Moreover, for ensuring the secure sailing, a multi-objective energy management model is established with the consideration of economic and environmental benefits. Meanwhile, the requirements of supply-demand balance, velocity, voltage security and so on are considered as well in the energy management. Then, a distributed energy management strategy based on ADMM algorithm is proposed. Finally, simulation results of a 5-node test system proves the effectiveness of the constructed energy management model and the distributed algorithm.

*Index Terms*—ship-integrated energy system, distributed optimization, energy management

## I. INTRODUCTION

With the deepening of economic and manufacturing cooperation among countries, we have gradually entered the era of globalization. As a major way of exchange of capital, goods, technology, and services among different countries and territories, the export value of international trade has been on the rise since 1950. Because of its low cost per unit of cargo delivery, wide route coverage, and other characteristics, the shipping industry has undertaken most of the global bulk cargo transport and has gradually become the most important manner of transport for trade exchanges between countries and supply chain operations between enterprises [1], [2]. As shown in Figure 1, between 1980 and 2022, the overall capacity of the shipping industry continued to rise, in which

This paper is supported by the National Natural Science Foundation of China (under Grants 52371360, 52201407, 51939001, 61976033) and the China Scholarship Council Program (under Grant 202406570011). *Corresponding author: Fei Teng*

cruise ships (tankers), container ships (containers), and cruise ships (cruises) capacity increased significantly, affected by the epidemic and other factors, general cargo (general cargo) capacity has slightly decreased [3]. By 2022, there are 10 ports in the world with a throughput of more than 15 million TEUs (Twenty-Foot Equivalent Unit, TEU), and the order of throughput from large to small is Shanghai (CNSHA), Singapore (SGSGP), Ningbo-Zhoushan ( CNZOS), Shenzhen (CNSZN), Qingdao (CNQIN), Guangzhou (CNGUA), Busan (CNBUS), Tianjin (CNTJN), Los Angeles/Long Beach (USLSA), Hong Kong (HKHKG).

However, the booming international trade and the revival of the shipping industry depend on huge fossil energy consumption, accompanied by the emission of various greenhouse gases (GHG) such as nitrogen and sulphur, which aggravate global warming and the melting of glaciers, and run counter to the concept of ecologically sustainable development [4]. In 2022, the transport sector will account for approximately 20% of global greenhouse gas emissions, second only to the electricity sector. As a necessary guarantee for international trade transactions, the greenhouse gas emissions of international shipping, international aviation and international rail account for 58.8%, 35.3% and 5.9% of the total international trade transport emissions, respectively. Carbon dioxide, as the main component of greenhouse gas emissions from shipping, accounts for more than 90 per cent of the total, and its total emissions have been on a sharp upward trend overall since 1970 until 2021. In 2021 alone, the shipping industry will emit about 700 million tonnes of carbon dioxide into the atmosphere, an increase of about 5% over the previous year. According to the International Maritime Organization (IMO), current $CO_2$ emissions from shipping industry have doubled since 1990 and reached a staggering 701.9 million

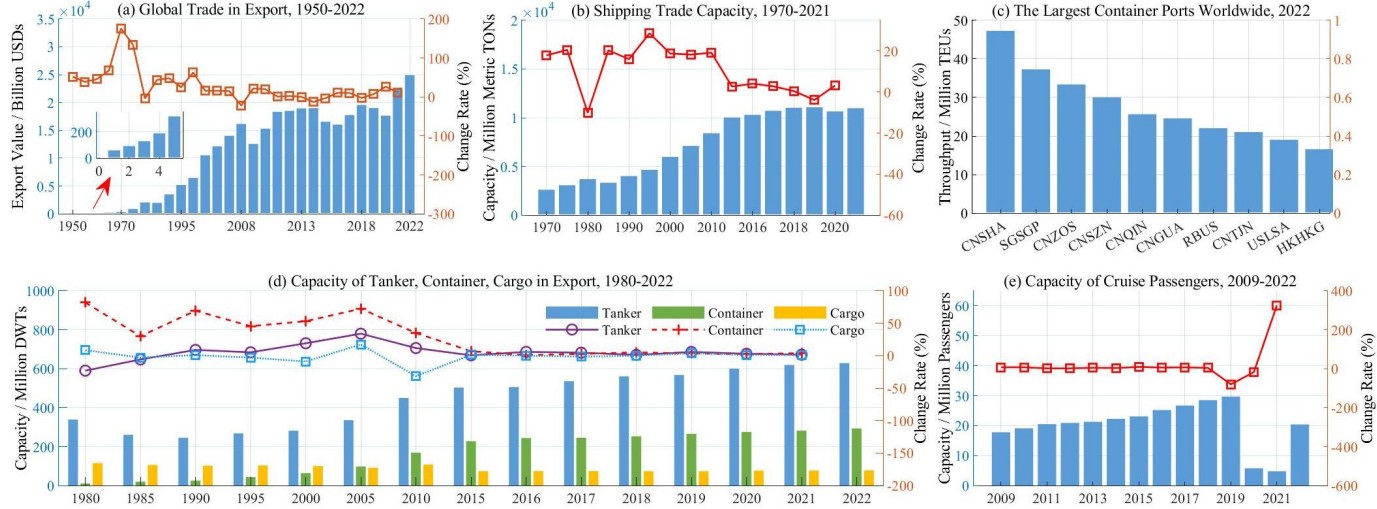

Fig. 1. Trends of Export Trade and International Maritime Trade

tonnes in 2017, as shown in Figure 2. If timely and effective improvement measures are not taken, the total amount of carbon dioxide generated by the shipping industry from fossil energy consumption will rapidly grow to 2.50-3.65 billion tonnes in 2050, accounting for about 18% of total global carbon emissions. To address the contradiction between the development of the maritime economy and the high level of carbon pollution from shipping, IMO and its subsidiary body, the Maritime Environment Protection Committee (MEPC), have established a number of regulations and strategic targets to reduce the total carbon emissions from the global shipping industry since 1997, as shown in Figure 3. In 2023, IMO member states unanimously agreed to adopt strategic carbon reduction targets that are expected to reduce emissions from international shipping by at least 20% and 70% relative to 2008 levels by 2030 and 2040, respectively, and to achieve zero GHG emissions by 2050. To this end, national classification societies and shipbuilding enterprises have paid extensive attention to the application of new energy in ship energy systems and invested in the research and development of on-board new energy equipment, thus promoting the innovation of ship energy systems.

According to the type of ship and its operating characteristics, the connection of energy devices in the current ship energy system with new energy devices and traditional fossil fuel devices can be broadly classified into three types, i.e., radiation pattern for small and medium-sized ships, ring pattern for large ships, and two-end pattern, which are shown in Fig. 4. Among them, radiation pattern is the most common and relatively mature technology, which is widely used. Compared with the radiation pattern, the ring pattern and the two-end pattern are equipped with a large number of non-professional energy supply equipment, the structure is relatively complex, and are mostly used in the energy network of large ships [5].

However, with the continuous progress and development of renewable technology, ship technology and information technology, more and more non-professional energy device and intelligent device integrated into the ship, the energy system presents a fully distributed flat structure coupled with a variety of heterogeneous energy sources. The cruise ship Ecoship, integrating diesel, natural gas, and photovoltaic panel, reduces carbon emissions by 40% compared with the average 60,000-tonne-class large cruise ship. Ship integrated energy system (S-IES) is a typical power-heating coupling energy system, which takes energy management system as the core and heterogeneous energy conversion centre as the hub, and uses both traditional energy devices and renewable energy device. It can reduce the dependence on traditional fossil fuels for sailing and improve the efficiency of energy utilization [6], which provides continuous and high-quality energy for the normal operation [7]. Therefore, S-IES and its energy management problem are gradually gaining wide attention in the related fields.

## II. SHIP-INTEGRATED ENERGY SYSTEM AND ITS ENERGY MANAGEMENT

### A. Structure and Framework for S-IES

As a core unit to ensure secure and reliable navigation of ships, S-IES provides continuous and high-quality energy for the energy system, communication and navigation system, and mechanical towing system. Ship energy management system carries out real-time monitoring and collection of shipboard load-side equipment, gathers communication and navigation systems as well as wind, wave, and other climate perturbation information to predict the shipload demand, and finally sends the load demand value to the energy supply side, and solves the optimal energy management scheme based on the intelligent algorithm to realize secure sailing in a long-distance voyage, and the structure of the system is shown in Fig.5.

Energy devices in S-IES can be roughly divided into five categories: power-only device (PO), heating-only device (HO), combined heating and power device (CHP), energy storage

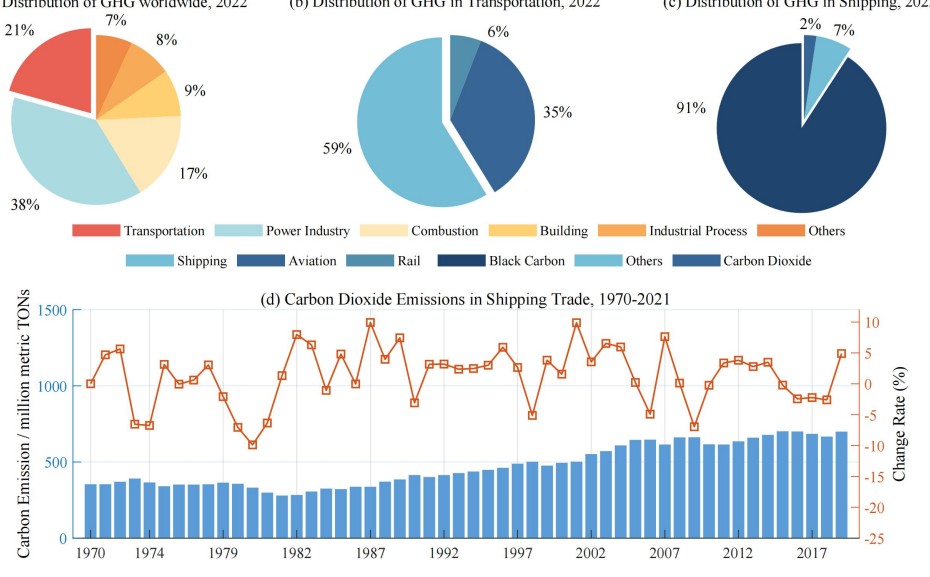

Fig. 2. Analysis of GHG Distribution Worldwide

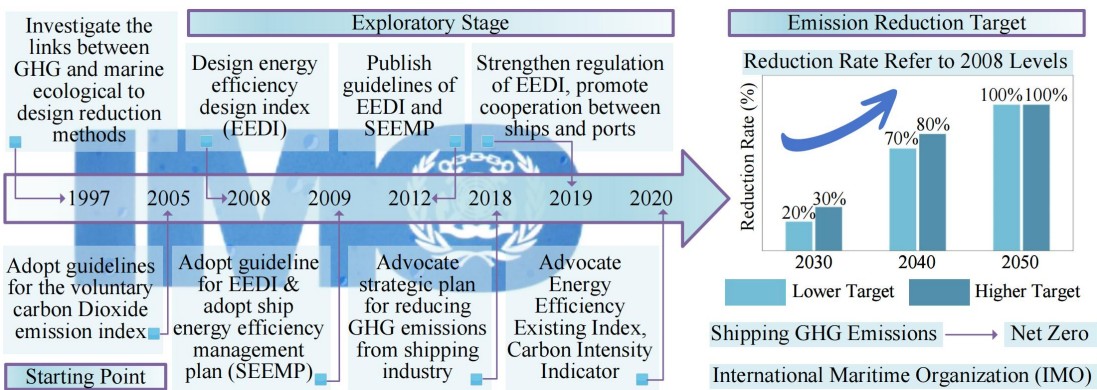

Fig. 3. Regulations development for GHG Emission Reduction

device (ESD) and load device (LD), storage device (ESD) and load device (LD).

For improving the energy supply performance of S-IES and ensuring that the energy supply side equipment provides continuous and high-quality energy for the safe navigation of the ship, it is necessary to carry out an in-depth analysis of the energy management problems of S-IES. The energy management system of S-IES is based on the theory of multi-intelligent body system, which realizes the bidirectional interaction of information and energy among the shipboard-distributed energy equipment. Then it formulates the optimal energy management strategy for the distributed energy management of S-IES. Ship energy efficiency monitoring technology, as a key role in ensuring the accuracy of the analysis of S-IES energy management issues, provides the necessary guarantee for maximizing the economic and environmental benefits of S-IES and has received extensive attention worldwide, as shown in Table I. Specifically, the EFMS designed by Ascenz Marorka, includes fuel pilferage prevention measures

and reporting capabilities for tracking fuel usage and analyzing data over time, which plays a vital role in saving operational costs by analyzing the information collected by the EFMS. For improving the operational fuel efficiency of ships, Germanischer Lloyd proposes the ECO-Assistant. It allows mariner to sail at optimal trim during voyage by acquiring the optimal trim angle in different sailing conditions. Moreover, the ECO-Assistant depends on the existing shipborne devices without installing extra modifications to the ships. By utilization of the routing algorithms, VVOS is constructed to optimize each route for on-time arrival while minimizing fuel consumption and avoiding weather damage. Additionally, the Electronic Chart Display and Information System (ECDIS) and Integrated Navigation System (INS) can receive the voyage scheme planed by VVOS to realizing secure check and execution. NAPA-VO is a software designed by NAPA, which realizes the improvement of operational efficiency. The created route with NAPA-VO concerns different fuel types and the corresponding features under emission control areas. A novel propulsion

TABLE I
ENERGY CONSUMPTION DETECTION TECHNOLOGY AROUND THE WORLD

| Instituation | Product | Functions and Characteristics |
|---|---|---|
| Ascenz Marorka | Electronic Fuel Monitoring System(EFMS) | Monitor real-time data on fuel usage. Develop energy management plan to reduce fuel consumption. |
| Germanischer Lloyd | ECO Assistant | Calculate optimum trim angle without making extra devices. Improve the operational fuel efficiency of ships. |
| Jeppesen Marine | Vessel and Voyage Optimization Services (VVOS) | Optimize route for on-time arrival. Voyage plan can be received by ECDIS and INS. |
| NAPA | NAPA Voyage Optimization (NAPA-VO) | Decrease operation cost by improved schedule adjustment procedure. Increase efficiency in altering the voyage plans. |
| ABB | ABB Dynafin | Cutting annual greenhouse emissions by at least 50% in future. Decrease the energy consumed by propulsion systems. |
| Hangzhou Yagena Technology Co., LTD. | Intelligent Ship System and Equipment (ISSE) | Calculate the carbon intensity index. Rank the carbon emission level. |

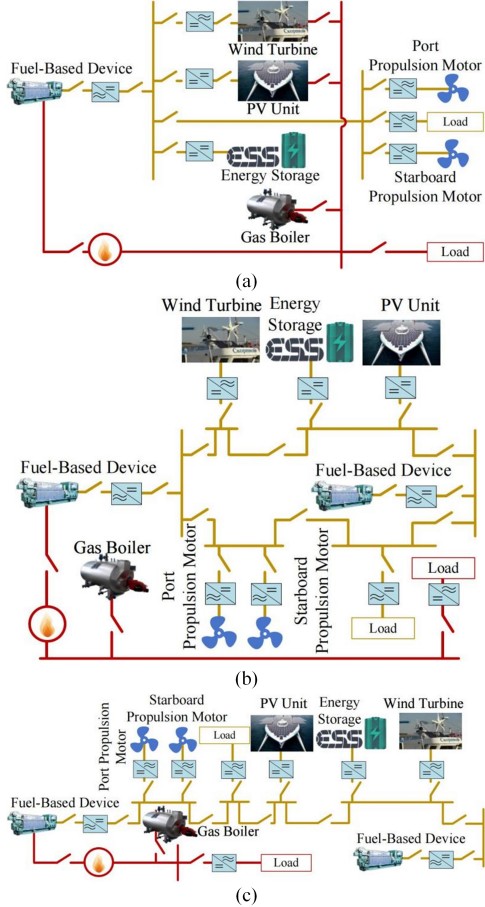

Fig. 4. (a) Structure of Radiation Pattern for S-IES, (b) Structure of Ring Pattern for S-IES, (c) Structure of Two-End Pattern for S-IES.

concept for ships is proposed by ABB Dynafin, which reduces greenhouse emissions by at least half. Moreover, compared to the traditional shaftline ships, the new technology is set to decrease propulsion energy consumption by up to 22%. ISSE can automatically calculate the carbon intensity index and related data sampling, which realizes the CII rating

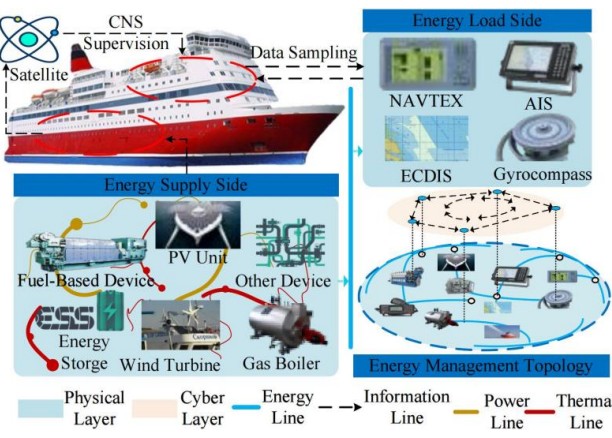

Fig. 5. Structure for S-IES

automatically and provides guarantees for sailing voyages.

At present, with the risen on awareness of sustainable development, the economic benefit $F_{EC}$ is not the only factor which should be concerned in the energy management of S-IES. For this reason, the environmental $F_{CA}$, and social $F_{SC}$ benefits are considered in this paper as well. Moreover, to ensure the high-security sailing and provide high-quality energy to the ships, the utilization of energy is also another major factor. Therefore, the objective function of the energy management problem for S-IES is constructed as,

$$\min\{F_{EC}, F_{CA}, F_{SC}\}. \tag{1}$$

Note that, the above objective function represents a compromise between economic benefits and carbon emissions, rather than indicating the smaller of $F_{EC}$, $F_{CA}$ and $F_{SC}$ as the objective function for S-IES energy management. In addition, physical constraints need to be imposed on ships and energy equipment to achieve secure performance during sailing voyage. Then, the energy management model of S-IES can be established as below.

$$\min\{F_{EC}, F_{CA}, F_{SC}\},$$
$$\text{s.t.} A(X) = 0, B(X) \leq 0, \tag{2}$$

where, $X$ is the decision variable of the energy management problem, which presents the energy output, $A(X)$ and $B(X)$ are equality and inequality constraints, respectively. Specifically, $A(X)$ contains supply-demand balance constraints, voltage security constraints. $B(X)$ includes energy output constraints, velocity constraints, which are shown below.

*1) Supply-demand balance constraints:*

$$\sum \left( p_i^{\text{fu}} + p_i^{\text{re}} + p_i^{\text{chp}} - p_i^{\text{ld}} \right) = 0 \tag{3}$$

$$\sum \left( h_i^{\text{fu}} + h_i^{\text{re}} + h_i^{\text{chp}} - h_i^{\text{ld}} \right) = 0 \tag{4}$$

where $p$ and $h$ are power and heating outputs, respectively. $i, l$ presents the node sequence of S-IES. fu, re, chp, ld are fuel-based generators, renewable-based generators, combining heating and power device, and load, respectively.

*2) Voltage security constraints:*

$$\sum \alpha_{i,l}^{\max} p_i + \Delta V_l^{\max} = \pi_l^{\max} \tag{5}$$

$$\sum \alpha_{i,l}^{\min} p_i + \Delta V_l^{\min} = \pi_l^{\min} \tag{6}$$

where $\alpha$ notes the voltage sensitivity coefficients, which can be calculated by a given transmission topology of S-IES. $\Delta V$ is the voltage excursion index, which indicates the voltage security margin. $\pi$ is a constant influencing by the reactive power. min and max are noted as the minimal and maximal values of the corresponding variables, respectively. Specifically, (5) and (6) present the upper and lower bounds of velocity security boundaries, respectively.

*3) Energy output constraints:*

$$p_i^{\min} \leq p_i \leq p^{\max}, h^{\min} \leq h_i \leq h^{\max} \tag{7}$$

*4) Velocity constraints:*

$$v^{\min} \leq v \leq v^{\max} \tag{8}$$

where $v$ describes the velocity of the ship.

### B. Communication Topology for S-IES

*1) Centralized Energy Management Strategy:* It relies on a centralized controller, which has significant advantages in terms of calculation rate and accuracy. The controller collects real-time information on the operation of all energy devices in the S-IES, and calculates the optimal energy management schemes based on the relevant data to complete the centralized control and management for the S-IES. However, the centralized strategy is prone to single point of failure, poor stability, limited scalability and huge investment costs.

*2) Decentralized Energy Management Strategy:* According to the operating mechanism and working status of the energy equipment, the ship can be divided into multiple compartments, and the decentralized energy management strategy places the controllers in the compartments. Thus the control and management of the local energy system can be realized. Moreover, the decentralized strategy does not require real-time communication between the controllers in each cabin, and the energy equipment distributes the energy by itself, so

the system response speed is fast and the scalability is strong. However, due to the lack of collaborative control between the controllers, the global optimization at the system level cannot be achieved. Therefore it is highly susceptible to external interference, which has a negative impact on secure navigation.

However, the flat distributed structure of the energy network of S-IES presents that the traditional centralized and decentralized energy management strategies are no longer applicable, and a fully distributed energy management strategy continues to be utilized to complete the search for the optimal solution in order to achieve the reliable operation of S-IES.

*3) Distributed Energy Management Strategy:* It relies on the communication network to achieve real-time information exchange between energy devices within the S-IES, so as to achieve complementary interconnection of information between neighbouring energy devices. When a local energy device fails to operate normally, the controllers of each energy device can achieve local interconnections and send signals to the controllers of neighbouring energy devices to ensure safe and stable operation of the S-IES based on the distributed communication network.

### C. Main Algorithm

To simplify the multi-objective functions of the energy management problems, the linear weighted sum method, the energy management model can be reconstructed as below.

$$\min \left\{ \beta_1 \cdot F_{\text{EC}} + \beta_2 \cdot F_{\text{CA}} + \beta_3 \cdot F_{\text{SC}} \right\},$$
$$\text{s.t.} A(X) = 0, B(X) \leq 0, \tag{9}$$

where $\beta_1$, $\beta_2$ and $\beta_3$ are constant parameters of $F_{\text{EC}}$, $F_{\text{CA}}$, and $F_{\text{SC}}$, respectively. Moreover, the related parameters satisfy $\beta_1 + \beta_2 + \beta_3 = 1$.

Define the communication topology of the constructed S-IES as $G = \{T, B, W\}$, where $T$, $B$, and $W$ are node set, connected edge set and connected weighted sets, respectively. Specifically, $T = [v_i | i \in \Omega]$ and $\Omega$ is noted as the energy device set. Moreover, $B \subseteq T \times T$ is related to the node set. $W = [w_{i,l} | i, l \in \Omega]$, where $w_{i,l}$ is the connected weighted parameter between the $i$th node and the $l$th node. Considering the different relationships between the $i$th node and the $l$th node, $w_{i,l}$ can be calculated as below.

$$w_{i,l} = \begin{cases} 1/(|N_{zi}| + |N_l| + \varepsilon), l \in N_i \\ 1 - \sum_{l \in N_i} 1/(|N_i| + |N_l| + \varepsilon), i = l \\ 0, otherwise \end{cases} \tag{10}$$

where $\varepsilon$ is a small positive constant. $N_i$ and $|N_i|$ are the neighbor set of the $i$th node and its cardinal number. Similarity, $N_l$ and $|N_l|$ are the neighbor set of the $l$th node and its cardinal number. When the connected weighted value between $i$th node and the $l$th node equals to 0, it means that the information exchange can not occur between the mentioned nodes.

Since the high calculation speed, accuracy, and reliable performed by the alternating direction method of multipliers (ADMM), a fully distributed energy management strategy for S-IES is designed in this paper. It realizes the bi-directional transmissions of energy and information, which improves the

reduction of communication resource and is suitable to the flat and distributed structure of S-IES. Then, the main algorithm can be designed as below.

*1) Iteration of energy output:*

$$X_{i,k+1} \in \arg\min\{f(X) + \psi + \lambda_{i,k}^{\mathrm{T}} W^{\mathrm{T}} A X_{i,k}\}, \quad (11)$$

where $\lambda$ is the incremental cost. $A$ describes the physical relationship among nodes of S-IES, which is defined by physical constraints. $k$ is the sequence of time slot.

*2) Iteration of output error:*

$$d_{i,k+1} = W d_{i,k} + A(X_{i,k+1} - X_{i,k}), \quad (12)$$

where $d$ is the output error of S-IES.

*3) Iteration of incremental cost:*

$$[\overset{\leftrightarrow}{\lambda}_{\Omega_{\mathrm{W}}^z,k+1}, \lambda_{z,k+1}]^{\mathrm{T}} = W[\overset{\leftrightarrow}{\lambda}_{\Omega_{\mathrm{W}}^z,k}, \lambda_{z,k}]^{\mathrm{T}} + \tau d_{z,k+1}. \quad (13)$$

Repeat the iterations of energy output, output error, and incremental cost until each variable converge a preset threshold.

## III. SIMULATION RESULTS

To verify the effectiveness of the designed algorithm, a 5-node S-IES is utilized as a test system, and the detailed physical/communication topology containing connected weighted parameters is shown in Fig.6. Moreover, in the test system, there are 2-fu, 1-re, 2-ld, and the operational coefficients and carbon emission parameters are presented in (14). Assume the load demand of power and heating are [4.5, 2.4](MW), respectively.

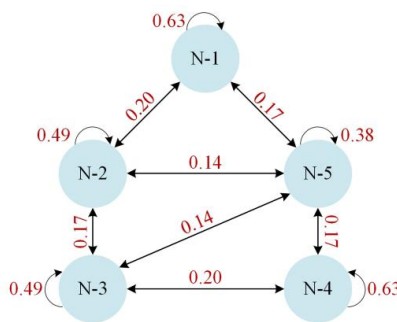

Fig. 6. Structure for the test S-IES

$$\begin{aligned}
C_1^{\mathrm{fu}} &= 0.040 * h^2 + 25 * h + 99 \\
C_1^{re} &= 0.043 * p^2 + 22 * p + 80 \\
C_2^{\mathrm{fu}} &= 0.035 * p^2 + 18 * p + 120 \\
C_1^{\mathrm{ld}} &= -0.013 * p^2 + 46 * p + 30 \\
C_2^{\mathrm{ld}} &= -0.015 * h^2 + 70 * h + 40 \\
E_1^{\mathrm{fu}} &= 0.0648 * h^2 - 2.7 * h + 41 \\
E_2^{\mathrm{fu}} &= 0.0520 * p^2 - 2.3 * p + 50
\end{aligned} \quad (14)$$

The trajectories of incremental costs of 5 nodes in the test S-IES are depicted in Fig.7. It can be found that the variable of incremental cost can be converged within 20 iteration steps. And the specified value of the final incremental cost is 0.3066 (p.u.). Moreover, the calculated optimized energy management solution is same as the solutions obtained by centralized strategy, which verifies the accuracy of the designed algorithm.

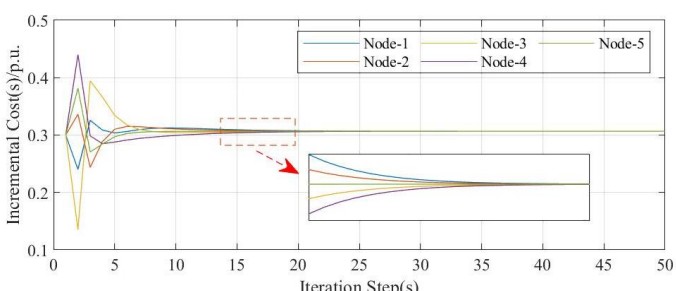

Fig. 7. Trajectories of incremental costs

## IV. CONCLUSION

In this paper, the development of S-IES has been analyzed. A multi-objective energy management model for S-IES has been constructed considering economic benefits and carbon emissions. Meanwhile, the entire sailing requirements for ships are considered in the construction of energy management model. Additionally, to search for the optimization solutions in a distributed manner, an energy management algorithm has been proposed based on ADMM theory. Simulation results proves the accuracy and effectiveness of the designed energy management model and the distributed algorithm.

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
