# OpenReview forum: "Distributed Energy Management for Ship-Integrated Energy System Considering Economic and Environmental Benefits"
_IEEE.org/ICIST/2024/Conference — IEEE ICIST 2024 Conference Submission_

### Official Review · Reviewer_7jbP · 2024-08-21
**Accept**

**Rating:** 7
**Confidence:** 4

**Review:**

This paper analyzes energy management in the shipping industry to reduce reliance on fuel-based resources. It reviews shipboard energy system development, establishes a multi-objective model considering economic and environmental benefits, and proposes a distributed ADMM-based strategy. Simulation results validate the model and algorithm's effectiveness in a 5-node test system.

How does the multi-objective energy management model address the trade-offs between economic and environmental benefits?

What are the comparative advantages and potential limitations of the ADMM-based distributed energy management strategy in contrast to centralized and other distributed approaches?

Can the document provide more insights into the development methods and parameter selection criteria for the shipboard integrated energy system's communication topology?

How does the algorithm's effectiveness scale with different sizes and types of shipboard integrated energy systems, and what are its generalizability considerations?

---

### Official Review · Reviewer_axve · 2024-08-21
**This paper studies the energy management problem is analyzed in this paper to decrease the dependency of fuel-based resources in the shipping industry. The feasibility of the designed control approach is proven via the simulation example. However, the following suggestions need to be considered in the revised manuscript to further improve the quality of this paper.**

**Rating:** 10
**Confidence:** 3

**Review:**

1. What are the innovations of the distributed energy management strategy based on the ADMM algorithm?
2. What are the significant advantages of this distributed energy management strategy based on the ADMM algorithm compared to existing energy management methods?
3. Please carefully check the grammar errors in this paper to ensure readability and fluency for the readers.

---

### Official Review · Reviewer_gn1g · 2024-08-22
**The paper is logically clear, and the simulation results are abundant. Recommended publication.**

**Rating:** 7
**Confidence:** 3

**Review:**

This paper studies the energy management problem is analyzed in this paper to decrease the dependency on fuel-based resources in the shipping industry. There are the following questions need to be considered:
1. Why do economic and environmental benefits have an impact on navigation safety? Please provide a brief explanation on this point. What role does the multi-objective energy management model play in it?
2. Please provide more details about the proposed distributed energy management strategy based on the ADMM algorithm. What are its main functions or advantages?
3. Some grammar issues should be checked and corrected.

---

### Decision · Program_Chairs · 2024-09-08

Accept (Oral)